# The War against Tuberculosis: A Review of Natural Compounds and Their Derivatives

**DOI:** 10.3390/molecules25133011

**Published:** 2020-06-30

**Authors:** Morgan Maiolini, Stacey Gause, Jerika Taylor, Tara Steakin, Ginger Shipp, Purushottam Lamichhane, Bhushan Deshmukh, Vaibhav Shinde, Anupam Bishayee, Rahul R. Deshmukh

**Affiliations:** 1School of Pharmacy, Lake Erie College of Osteopathic Medicine, Bradenton, FL 34211, USA; MMaiolini21274@rx.lecom.edu (M.M.); SGause89417@rx.lecom.edu (S.G.); jtaylor20502@rx.lecom.edu (J.T.); TSteakin48474@rx.lecom.edu (T.S.); 2Lake Erie College of Osteopathic Medicine, Bradenton, FL 34211, USA; GShipp@lecom.edu; 3School of Dental Medicine, Lake Erie College of Osteopathic Medicine, Bradenton, FL 34211, USA; plamichhane@lecom.edu; 4Department of Chemistry, Kavayitri Bahinabai Chaudhari North Maharashtra University, Jalgaon 425 001, Maharashtra, India; bhushanddeshmukh@gmail.com; 5Department of Pharmacognosy, Poona College of Pharmacy, Bharati Vidyapeeth (Deemed to be University), Pune-411 038, Maharashtra, India; vaibhavshinde847@gmail.com

**Keywords:** Tuberculosis, *Mycobacterium tuberculosis*, multidrug resistance, antibiotics, natural compounds

## Abstract

Tuberculosis (TB), caused by the bacterial organism *Mycobacterium*
*tuberculosis*, pose a major threat to public health, especially in middle and low-income countries. Worldwide in 2018, approximately 10 million new cases of TB were reported to the World Health Organization (WHO). There are a limited number of medications available to treat TB; additionally, multi-drug resistant TB and extensively-drug resistant TB strains are becoming more prevalent. As a result of various factors, such as increased costs of developing new medications and adverse side effects from current medications, researchers continue to evaluate natural compounds for additional treatment options. These substances have the potential to target bacterial cell structures and may contribute to successful treatment. For example, a study reported that green and black tea, which contains epigallocatechin gallate (a phenolic antioxidant), may decrease the risk of contracting TB in experimental subjects; cumin (a seed from the parsley plant) has been demonstrated to improve the bioavailability of rifampicin, an important anti-TB medication, and propolis (a natural substance produced by honeybees) has been shown to improve the binding affinity of anti-TB medications to bacterial cell structures. In this article, we review the opportunistic pathogen *M. tuberculosis*, various potential therapeutic targets, available therapies, and natural compounds that may have anti-TB properties. In conclusion, different natural compounds alone as well as in combination with already approved medication regimens should continue to be investigated as treatment options for TB.

## 1. Introduction

*Mycobacterium tuberculosis* is one of the world’s leading causes of infectious disease-related death despite 90 years of vaccination and 60 years of chemotherapy [1]. *M. tuberculosis* is a non-motile, acid-fast and an obligate aerobe (requires oxygen to grow). These organisms are bacillus-shaped (measuring 0.22–0.6 × 10^10^ μm) and cell growth is slow with cell doubling time ranging from 2 h to more than 20 h [2]. The complexity of the mycobacterial cell wall is a major distinctive feature that sets *Mycobacterium* apart from other bacterial organisms [3]. The mycobacterial cell envelope is critical for cell function because many crucial processes are located in this structure [4]. Important functions of the cell wall include the protection of the bacterial cell from hostile environments, general antimicrobial resistance, the transport of nutrients as well as adhesion to host cell receptors [4]. A key component of the cell wall is mycolic acid which is not clearly differentiated by gram-positive or gram-negative staining; hence these organisms are referred to as “acid fast” [5,6].

TB is notable among infectious diseases because it is transmitted almost exclusively through the air [7]. An important step in spreading TB infection is the ejection of bacilli from the lungs of infected persons and release into the surrounding environment [7]. The organisms are then inhaled by susceptible individuals - typically those who are close contacts and immunocompromised persons [8,9]. The organisms later proliferate in the lungs, stimulating the immune system of the host, specifically the activation of phagocytes [10]. This results in a mass referred to as a granuloma, which consists of necrotic lung tissue and immune cells such as T cells, B cells and macrophages [10]. The granuloma is the pathologic hallmark of the host response to infection with TB [11]. About 10% of all persons initially infected progress to active disease which is symptomatic and transmissible. However, most people (90%) develop a latent (or dormant) infection, which is asymptomatic and non-transmissible [12]. Latent tuberculosis infection is characterized by the presence of immune responses to TB infection without clinical evidence of active TB [13]. Persons with latent TB are at risk for progression to active disease (referred to as “reactivation”) [14,15]. Individuals with active TB (the spreading of bacilli in the lungs and beyond the lungs in some patients) experience symptoms, such as decreased appetite, fever, tiredness, and weight loss [15]. Persons with pulmonary disease often have a persistent cough; those with advanced pulmonary disease typically exhibit coughing up blood (hemoptysis) [15]. Reactivation TB occurs when those with a latent TB infection progress to active infection; the organism can persist for decades within a granuloma structure and as a result of an immune-compromising event [e.g., diabetes, human immunodeficiency virus (HIV) infection, malnutrition, and cancer], the bacteria can reactivate [10].

The diagnosis of TB can be challenging especially in resource-limited countries [16]. The tuberculin skin test (TST) is the best-known method for detecting exposure to TB. TST detects cell-mediated immunity to TB through a delayed-type hypersensitivity reaction using a protein precipitate of heat-inactivated tubercle bacilli (purified protein derivative tuberculin). The TST has been the standard method of diagnosing latent TB infections [17]. There are a number of techniques available to identify TB in the laboratory that are beyond the scope of this work and are described in detail elsewhere [16,17,18,19], however, the most common methods will be mentioned in this review. Phenotypic laboratory methods include acid fast smear microscopy (identification of sputum samples using a conventional light microscope and Ziehl-Neelsen staining); the microscopic observed drug susceptibility procedure (growing organisms in liquid culture, concentrating organisms into a “pellet”, placing the pellet on media containing antibiotics and utilizing an inverted light microscope to detect susceptible and resistant TB organisms); and agar proportion testing (using drug-free and drug-containing solid media to detect organisms as well as possible resistance patterns) [16]. Genotypic (molecular) methods include the line probe assay which utilize polymerase chain reaction (PCR) and reverse hybridization with specific probes to identify the RNA polymerase B gene in resistant organisms and the Xpert© MTB/RIF assay (Cepheid Incorporated, Sunnyvale, CA, USA) which utilizes real-time PCR technology to concurrently diagnose TB and detect rifampin (RIF) resistance [20,21].

In 2018, *M. tuberculosis* infected over 10 million people, killing 1.5 million persons [22]. One-third of new cases remain unknown and many individuals do not receive proper treatment [1]. Multidrug-resistant (MDR) TB remains a significant threat to society, especially to those with underdeveloped health care infrastructures [1,18]. The WHO estimates that there were 600,000 new cases of TB that exhibited resistance to rifampicin (RIF), one of the most effective first-line drugs to combat TB. “First line” medications are defined as those that the WHO recognizes as standard therapy for TB [23]. TB generally is a curable disease; however, there is concern about rising antimicrobial resistance throughout the world [24,25,26]. Active TB is usually treated with a 6-month course of RIF, isoniazid (INH), pyrazinamide (PZA) and ethambutol (EMB), also referred to as R.I.P.E. therapy.

According to the WHO, 484,000 patients with TB had drug resistant TB [22,27]. Drug-resistant TB impedes progress towards control of the disease [28]. Multi-drug resistant (MDR) organisms are defined as those that are resistant to first-line antibiotic medications for TB, such as INH and RIF [22,27]. An important factor in multidrug resistance is the lack of compliance to TB treatment [28]. As noted, multiple drugs are required to treat TB disease and these extended drug regimens are challenging to both patients and health care systems [1]. Extensively drug-resistant (XDR) TB has been found in 127 countries [27,28]. XDR-TB is defined as bacterial organisms that are resistant to any fluoroquinolone and at least one second line (injectable) drug, such as capreomycin, amikacin, and kanamycin [29]. XDR-TB is responsible for greater than 10 % of MDR-TB cases in the Russia Federation and South Africa, where the impact has been devastating due to frequent co-infection with HIV [28]. Treatment of MDR-TB is an extended process, ranging from 20-30 months of treatment [30]. If TB is resistant to at least one of the first-line treatments (traditionally RIF or INH), second-line therapy treatments are incorporated into the treatment regime. Medications that are utilized to treat TB (including MDR-TB) are detailed in Table 1. In cases of XDR-TB, different combinations of medications from MDR options are used along with various “off-label” medications (defined as an approved drug used for conditions not sanctioned by governmental regulatory agencies). Select off-label medications for TB include bedaquiline, linezolid, amoxicillin/clavulanate, meropenem, and thioacetazone [30]. Natural products have demonstrated different effects on bacterial organisms. Thus, many have been used in traditional medicine and drug development for pathological conditions, such as fever, tumors, and kidney disease. In an article by Salomon and Schmidt [31], different natural products were reviewed. They focused on the structures and possible effects on different microbes. For example, flavonoids have an inhibitory effect on enzymes in fatty acid synthesis in microbes. This was also shown in the effects of β-ketoacyl reductase in *Escherichia coli,* and β-hydroxyacyl ACP dehydratase in *Plasmodium falciparum* [31]. In another study, the antibacterial effects of natural compounds were assessed against Gram-negative organisms (such as *Escherichia coli and Klebsiella pneumoniae*). The 19 compounds that were examined demonstrated some antibacterial activity [32].

## 2. Therapeutic Targets

Various molecular targets against *M. tuberculosis* exist; those targets include DNA replication, RNA synthesis, energy metabolism, folate metabolism, and bacterial cell wall synthesis [33]. Mycobacterial cell wall inhibitors impede the synthesis of mycolic acids, arabinogalactan and peptidoglycan, important components of the mycobacterial cell wall [34]. These antibiotics corrupt the structural integrity of the bacterial cell envelope and this is believed to cause drug-mediated cell death. Targeting of DNA replication can be problematic due to many proteins that are involved in this process [35]. Structural targets involved in DNA replication include mycobacterial polymerase III holoenzymes (involved in the polymerization or elongation of DNA); DNA gyrases (important in DNA winding and unwinding during replication) and DNA replication initiators (structures important in the start of DNA replication) [35]. Medications that target RNA synthesis include those that inhibit bacterial DNA-dependent RNA polymerase formation, which are significant enzymes important in the synthesis of RNA [36]. All bacteria require energy to function. In aerobic organisms, the process of oxidative phosphorylation (the conversion of chemical energy from diverse sources of ATP) can be altered in *M. tuberculosis* to interfere with bacterial metabolism [33]. Folate or vitamin B9 is very important in prokaryotes, including *Mycobacterium*. Folate serve as essential cofactors in the synthesis of amino acids methionine, N-formylmethionyl-tRNA (a derivative of methionine) glycine, serine, purine, and thymidine [37]. Folates are also important in the formation of pantothenate (also referred to as vitamin B5) [37]. Pantothenate is an essential nutrient required for energy metabolism. It also functions as a cofactor in the synthesis of coenzyme A, which is important for fatty acids metabolism and the tricarboxylic acid (TCA) cycle [38].

Based on their mechanism of action, important anti-TB drugs are classified as inhibitors of electron transport across bacteria membranes (PZA); inhibitors of nucleic acid synthesis (RIF and quinolones) inhibitors of protein synthesis (aminoglycosides); and inhibitors of cell wall synthesis (INZ, ethambutol and ethionamide) [39,40]. Increasing microbial resistance to these agents has resulted in public-private partnerships to develop new treatments for TB; these partnerships include coordinated efforts between academia, clinicians, non-governmental organizations and the pharmaceutical industry [40,41]. New antimicrobial therapies include moxifloxacin (a fluoroquinolone), SQ109 (an ethylenediamine) and bedaquiline (TMC 207), a diarylquinolone [41]. Diarylquinolones are synthetic products that were derived from natural quinolones [41,42,43,44]. They interfere with mycobacterial energy production by binding and inhibiting F_1_F_0_ ATP synthase, an enzyme that creates the energy storage molecule ATP [45]. The disruption of energy production results in ATP depletion and imbalance in pH homeostasis, resulting in cell death [46]. In Figure 1, anti-TB drugs and their targets are shown.

## 3. Treatment

The major purposes of anti-TB therapy are to eliminate the *M. tuberculosis* bacillus as rapidly as possible; prevent drug resistance; and to avoid relapses by eliminating persistent bacilli [47]. TB treatment is one that requires both diligence and commitment, as it involves the combination of several different drugs over a lengthy period. The time to treat active TB is at least 6 months without complications and can last much longer (beyond 8 months) in cases with extensive resistance. These factors make adherence a topic for concern [48,49,50]. Adherence refers to the extent of which a patient’s history of therapeutic drug-taking coincides with the prescribed treatment [50]. There are many therapeutic combination options; for successful treatment, it is necessary to prescribe anti-TB drugs with an adequate dosage, for a specific time of exposure and whose effectiveness has been proved in in vitro tests (i.e., drug-susceptibility testing) [51]. To avoid the emergence of resistant TB strains, it is necessary to prescribe at least two effective drugs [51]. Currently, the United States Food and Drug Administration (FDA) has approved 10 drugs for the treatment of TB [48]. Included in those medications are certain drugs that are considered important for first-line treatment, i.e., R.I.P.E.) therapy [48]. The medications for R.I.P.E. therapy are used in varying combinations and are believed to be the most effective treatment option [48]. The challenge of TB therapy is to devise well-tolerated, effective, short-duration regimens that can be used successfully against susceptible and resistant TB in a diverse population of patients [52].

### 3.1. Newly Diagnosed Patients

Risk factors for contracting TB include employment in health care facilities; being a resident or employee in institutional settings; recent immigration within the past five years from high-incidence countries, and illicit drug use [53]. For newly diagnosed persons with drug-susceptible TB, the most effective treatment recommended by the WHO is 6 months of R.I.P.E therapy [48]. This regimen is divided into two phases: intensive and continuous. The purpose of the intensive phase is to kill the bacilli; the patient consumes one tablet of RIF, INH, PZA, and EMB every day for a span of 2 months [48]. The continuous phase’s purpose is to prevent reoccurrence; the patient ingests INH and RIF once every day for at least 4 months [48,54]. For optimal treatment, health care professionals will implement direct observational therapy (DOT or DOTS). In the DOTS approach, anti-TB medications are ingested by patients under the supervision of a healthcare worker ensuring that proper medications are given at appropriate intervals and at the correct doses [55]. Short-course DOTS restrictions, as well as frequency and length of dosing may vary in patients with HIV and patients with other serious illnesses, such as liver disease and renal insufficiency [48].

### 3.2. Multidrug-Resistant (MDR) and Extensively Drug-Resistant (XDR) Tuberculosis

Treatment of MDR-TB can be extremely challenging. Outbreaks of MDR-TB were originally thought to be driven by nosocomial (or hospital-acquired) transmission, particularly among HIV-positive patients [56]. However, researchers discovered the causes of the global spread of MDR-TB include ineffective and/or expensive treatment, facility-based transmission (such as in prisons) and ironically, development of resistance due to improperly administered DOTS therapy [56]. Improperly administered DOTS therapy renders TB organisms resistant due to repeated exposure to antibiotic agents [56]. Risk factors for MDR-TB include living in countries with high prevalence of MDR-TB HIV coinfection, the failure to respond to a first-line DOTS regimen; and relapse after a full-course treatment with a first-line regimen [56].

### 3.3. Important Factors in M. tuberculosis Infections and Limitations of Treatment

Important factors to consider when reducing TB infections include bacterial load and disease sites in infected patients; the nutritional/immunity status of both TB patients and the general public; screening activities; availability of treatment programs and compliance/adherence to treatment regimens [57,58,59]. Patients with extensive cavitary lung disease and high bacterial loads are highly infectious and remain so for a longer period of time [59]. In addition, persons with poor nutritional and immune status are more likely to be infected with TB [60]. According to the WHO, the primary objectives of screening are to ensure that active TB is detected early in order to reduce the risk of poor disease outcomes; to be aware of the adverse social and economic consequences of the disease; and to help reduce TB transmission [61]. For example, screening activities can be systematic (evaluating whole populations of people) or targeted at select risk groups such as intravenous drug abusers and recent immigrants from nations with a high incidence of TB [61]. Screening can also be passive (such as asking all people seeking health care about possible TB symptoms) [61]. Traditionally, treatment programs have focused on providing free medical services, especially in high-burden countries [62]. However, there are countries that require partial or total payment of medical service, including the treatment for TB [63].

In the United States, the control of TB infections is a multifaceted process involving employees in the private health care sector; federal agencies, such as the Centers for Disease Control and Prevention’s (CDC) and well as state and local health departments [64,65]. It is very important to involve local health departments as 77% of US-diagnosed TB patients receive most or all of their clinical care through these facilities [66]. One goal of these organizations is to reduce morbidity and mortality by preventing transmission of TB from contagious persons to the uninfected. These organizations also treat TB-infected patients by focusing on preventing the progression from latent TB to active TB disease [67]. The role of local public health case managers, community health nurses and clinicians trained in infectious disease control are crucial in activities, including case investigations, providing medical treatment to patients and educating patients and their families about the disease [67].

The limitations of conventional TB treatment involve many related factors, including non-adherence to treatment and non-compliance due to issues such as the length of treatment, abundance of medications taken and their side effects [48]. At minimum, patients require a 6-month commitment to their treatment. However, many patients improve medically after a few months of treatment and may believe that continuing their medications is no longer necessary [68]. In addition, irregular treatment (due to issues such as busy work schedules and lack of transportation to medical facilities) and poor quality of DOTS therapy are often cited as significant factors in non-adherence [69]. For many, there is also a stigma in being treated for TB; some patients believe that they have little social and psychological support and may stop treatment [70]. Moreover, medications used to treat TB may cause troublesome side effects that can negatively affect the patient’s quality of life and cause patients to stop taking their medications [71]; side effects include gastrointestinal distress, nephrotoxicity, hypothyroidism, neurotoxic effects (including depression) and ototoxicity (damage to cranial nerve VIII, resulting in hearing loss, and balance disorders) [56,72]. There has also been reports of patients sharing TB medications with other patients and “pill dumping” (purposely discarding required medications prior to medical appointments) [57]. In the US, there is legal pressure for many who are deemed willfully noncompliant. In extreme situations, those patients may be subject to legal intervention in the form of court-ordered medical care and completion of therapy as well as criminal confinement for completion of tuberculosis treatment when less restrictive measures have been unsuccessful [73]. Health care professionals have implemented a number to solutions to decrease non-adherence including laboratory assays to detect medication metabolites in the urine of patients; recruiting “treatment supporters” (defined as family and close friends) in dispensing medications to patients [57]; and the use of Video-Directly Observed Therapy, a smartphone-based approach that allows for remote treatment monitoring through patient-recorded videos [74].

## 4. Natural Compounds

The significance of natural products in antibacterial drug treatment has been indisputable. Historically, natural products have been important in therapy against TB [75]. For example, the peptide actinomycin (derived from the bacterial organism *Streptomyces antibioticus*) was the first described natural product that inhibited in vitro growth of *M. tuberculosis* [75]. The rifamycins (a group of antibiotics produced by the microorganism *Amycolatopsis mediterranei*) are an important class of antimicrobial agents [76]; RIF is the basis of treatment regimens for patients diagnosed with active tuberculosis [76]. Many natural products are biologically active and have better pharmacokinetic properties (e.g., superior absorption, distribution, and excretion) when compared to synthetic compounds [75]. Despite some of the issues in the field of natural product discovery (including *M. tuberculosis*’ unique cell wall, complex biosynthetic pathways and reduced solubility of some compounds) the fields of biology, pharmacology and medicinal chemistry have great potential to produce natural product-based therapeutics for TB treatment [29].

Natural compounds form the basis for many commonly used medications. This was recognized worldwide in 2015 when the Nobel Prize in Physiology or Medicine was awarded to Chinese research scientist Ty Youyou for her discovery of the herbal antimalarial compound artemisinin [77].Artemisinin (acquired from the wormwood plant *Artemisia annua*) is referred to as “qinghao” in China and has been utilized by herbal medicine practitioners for at least 2000 years for treatment against malaria [78]. In 1972, the active ingredient in the wormwood plant was purified and later renamed artemisinin; artemisinin is classified as a lactone endoperoxide and works by forming free radicals in blood, thereby inhibiting malaria parasite replication [79].

Concurrently, there has been a renewed interest in utilizing natural compounds as adjuvants in TB therapy including marine natural products [80]; “langdu” or traditional Chinese root [81] and garlic [82]. Immunoxel (Dzherelo), a water–alcohol extract of medicinal plants is often used as adjunct immunotherapy for TB and TB/HIV co-infection [24]. This was determined during clinical trials conducted by the multinational corporation Immunitor, LLC (Canada and Mongolia) and the governments of Ukraine, Mongolia, Canada and South Africa [83]. Consumption of medication took place in several oral forms, including pills, lozenges, and candies. The clinical trials determined that Immunoxel therapy (in combination with conventional treatments) resulted in increased weight gain and reduced systemic inflammation in patients. However, researchers also noted that these results occurred in patients that also took the anti-TB drugs alone [82].

In the remainder of this review, we discuss natural products that may display antimicrobial activity against TB, including turmeric (*Curcuma longa*), a perennial plant from the ginger family, and phloretin, an antioxidant which is found in *Malus domestica* (apple tree) leaves. Additionally, compounds, such as lactacystin produced by the microorganism *Streptomyces lactacystinaeus* and peptide aldehydes isolated from the marine fungus *Penicillium* have been investigated to determine their activities against TB infection. A summary of natural compounds listed in this article are featured in Table 2.

### 4.1. Curcumin

Curcuminoids, classified as natural polyphenol compounds, are derived from turmeric, a member of the ginger family (Zingiberaceae) [106]. Among them, curcumin, which is yellow-colored, is the primary component. Historically, it has been used in food, as a coloring agent, and in traditional medicine [106]. For centuries, curcumin has been used as an herbal medication for the treatment of various ailments including diabetic ulcers, rheumatism and anorexia [107]. Investigators have also considered curcumin as a possible treatment for tuberculosis. In one study, curcumin reduced the burden of intracellular tuberculosis in an immortalized monocytic human cell line (THP-1) [84,85]. Investigators pre-incubated THP-1 cells with increasing amount of curcumin (10, 30 or 50 µM) or 0.05% dimethyl sulfoxide (DMSO), a solvent used as a control for 1 h. The cells were later infected with *M. tuberculosis* (H37Rv strain). THP-1 cells were later lysed at 1 h (day 0), day 2 and day 4 post-infection. Cell lysates were analyzed by Western Blot (protein analysis), immunochemistry (via staining and fluorescent microscopy) and the nuclear factor-κB (NF-κB) p65 activation assay (which detects transcription factor activation during DNA replication) [84]. For control cells (THP-1 cells incubated with 0.05% DMSO), there were significant increases in the number of intracellular TB cells from 1 h (day 0) to 4 days after infection. However, in the presence of 10 µM curcumin, there was a trend toward inhibition of intracellular TB growth by 4 days. With 30 or 50 µM curcumin, there was a significant decrease in the number of intracellular TB cells detected at day 2 and day 4 when compared to control cells [84]. Curcumin also induced apoptosis of THP-1 cells. This was determined by terminal deoxynucleotidyl transferase dUTP nick end labeling staining (which detects DNA breakage during the final phase of apoptosis); Western blotting; and quantifying caspase-3 protein activation (proteins initiated during cell apoptosis). As the concentration of curcumin increased, there was also an increase caspase-3 protein and THP-1 apoptosis [84]. During this study, scientists noted that the important issues that must be addressed before conducting human in vivo studies include poor oral bioavailability, poor gastrointestinal (GI) absorption, as well as rapid metabolism and elimination [84].

A second study addressed the bioavailability issue by developing a nanoformulation for curcumin. Purified curcumin was repeatedly homogenized with ethanol and citric acid; the resulting slurry was dried at 60 °C in order to obtain a nano-curcumin powder [86]. Experimental mice were divided into 3 groups. One of the following treatments was given to each group; intraperitoneal injections of natural curcumin; curcumin nanoparticles that were suspended in phosphate buffered saline solution; or no treatment (control) [86]. Select mice were also fed the anti-TB drug INH *ad libitum* in their water supply. Blood samples were subsequently analyzed throughout the experiment and the animals were later sacrificed. The following analyses were completed on spleen and lung samples; in vitro and in vivo T-cell proliferation (lungs and spleen), hepatotoxicity analysis (liver), colony forming unit estimation of TB cells on 7H11 Middlebrook agar, Difco™ (spleen and liver); and flow cytometry assays (spleen and lungs) [86,108]. Curcumin nanoparticles significantly decreased hepatotoxicity with INH; additionally, increased T-cell-mediated immunity was detected among mice treated with curcumin nanoparticles when compared to mice given natural curcumin or with control mice [86]. In addition, the duration of antibiotic treatment needed for mice was reduced. Researchers summarized that nanoparticles of curcumin may be a favorable adjuvant therapy with standard tuberculosis regimen-and may possibly reduce the risk of MDR-TB and XDR-TB [86].

### 4.2. Phloretin

Flavonoids are polyphenolic compounds that can be found in fruits, vegetables, legumes, nuts and plant-derived foods (including coffee and red wine) [87]. In previous work, the polyphenolic compound phloretin, a dietary flavone found in apples (*M. domestica*) was discovered to have many biological functions, including possible antineoplastic and anti-inflammatory effects (such as production of chemokines, cytokines and other factors induced by leukocytes) [87]. Investigators conducted experiments to determine if phloretin also had antimicrobial effects on various strains of TB (H37Rv, MDR-TB and XDR-TB). Tuberculosis causes lung inflammation that contributes to disease pathogenesis; therefore, the effects of phloretin in interferon-γ-stimulated MRC-5 human lung fibroblasts (the principal cell of connective tissue) and lipopolysaccharide-stimulated dendritic (immune) cells were investigated [87]. First, culture broth of *M. tuberculosis* (H37Rv, MDR and XDR isolates) were grown and plated as previously described [87]. Bacterial organisms were then loaded on 96-well microfluidic agarose chips containing phloretin, INH (positive control) and Middlebrook 7H9 broth with 10% oleic albumin dextrose catalase. The plates were later incubated at 36 °C. Microfluidic agarose chips were then viewed with an inverted microscope (on day 3, 5, 7 and 9) [87]. The following laboratory techniques were also utilized in this study; Reverse-Transcription Polymerase Chain Reaction (RT-PCR), Western Blot, Enzyme-Linked Immunosorbent Assay (ELISA) and molecular docking/fluorescence quenching studies [87]. Phloretin inhibited growth of *M. tuberculosis* H37Rv, MDR and XDR-TB isolates. The study also demonstrated that phloretin exhibited safe and effective properties for reducing tuberculosis and associated lung inflammation by decreasing the effects of tumor necrosis factor-α (TNF-α) and interleukin-β [87].

### 4.3. Quercetin

Flavanols, the most abundant of the flavonoid molecules, are widely distributed in plants. Quercetin belongs to this class of phytochemicals [109]. Quercetin is found in many foods, including apples, berries, capers, grapes, and tea [109]. It is also the subject of studies investigating a wide array of targets against *M. tuberculosis*. In one such study, quercetin was utilized to target glutamine synthetase in three strains of *Mycobacteria* (H37Rv, smegmatis and phlei); glutamine synthase is an essential enzyme in nitrogen metabolism [88]. Activity of quercetin was detected using agar diffusion, broth microdilution, sulforhodamine B colorimetric assays and in vitro as well as molecular docking studies. Quercetin was found to inhibit all three strains of *M. tuberculosis*; the inhibition was directly proportional to quercetin concentration [89]. In another study, quercetin was used to target isocitrate lyase, an enzyme responsible for *M. tuberculosis* binding to host cells [89]. Isocitrate lyase is essential for the function of the TCA cycle and subsequent cell growth [110]. Fluorescence quenching, Lineweaver-Burke plots, pocket detection algorithm and blind docking methodologies were used in this study. Quercetin was discovered to decrease *M. tuberculosis* isocitrate lyase, resulting in an increased inhibitory effect on *M. tuberculosis* metabolism [89]. Additionally, quercetin at 3.57 µM (IC_50_) inhibited 25% of glucose metabolizing bacilli. Because of poor solubility of the compound above IC_50_, complete inhibition could not be achieved [90]. In a third study, quercetin, pomegranate juice and pomegranate peel extracts were used to target drug resistant *M. tuberculosis*. Kirby-Bauer disk diffusion, broth microdilution and tetrazolium microplate assay with 3-(4,5-dimethylthiazol-2-yl)-2,5-diphenyltetrazolium bromide (MTT) were employed to identify antimicrobial properties. MTT-based assays are widely used to evaluate microbial physiological state-including evaluation of viability and growth [108]. Pomegranate peel extracts displayed greater antimycobacterial activity than pomegranate juice and exhibited greater antibacterial and antitubercular potencies [91]. In an animal study, researchers utilized quercetin and polyvinylpyrrolidone to target tissue necrosis in the internal organs of mice caused by *M. tuberculosis*. Macroscopic and microscopic investigations of lesions were conducted. Mice treated with quercetin demonstrated dramatically reduced lesions; quercetin also prevented necrosis from spreading to other tissues when combined with anti-tuberculosis medications. This study suggests that quercetin and antibacterials have hepatoprotective effects against *M. tuberculosis*. Hepatic tuberculosis (or TB of the liver) is a rare extrapulmonary manifestation of the disease in which the organism spreads beyond the lungs to the liver [92]. Hepatic TB disease is increasingly found in patients co-infected with HIV; public health workers in TB-endemic regions should suspect hepatic TB in patients with a fever, respiratory symptoms, enlarged liver, and elevated liver enzymes [92].

### 4.4. Tannins

Tannins (also referred to as tannic acid) are water-soluble polyphenols that are present in many plants, including barley, plum, and strawberries [111]. Tannins, obtained from extracts of a perennial flowering plant (*Globularia alypum* L.), were investigated for their antioxidant and antituberculosis activity. Extracts of *G. alypum* were obtained by two extraction methods; the first method utilized a methanol, petroleum and dichloromethane mixture; and a second method used a methanol, acetone and water mixture [94]. Both mixtures were later evaluated for the presence of tannins, flavonoids, and anthocyanins; both tannins and anthocyanins are classes of flavonoids [47]. Analysis of the mixtures were conducted using α-diphenyl-β-picrylhydrazyl (DPPH) and 2,29-azinobis-3-ethylbenzothiazoline-6-sulfonic acid (ABTS) assays [95,96]. Both assays measured the antioxidant capacities of natural products; these spectrophotometric techniques demonstrate the radical scavenging ability of antioxidants, even when present in complex biological mixtures such as food extracts and plants [97]. After analysis by DPPH and ABTS, both *G. alypum* mixtures were evaluated for activity against *M. tuberculosis* strain H37RV by using a colorimetric micro assay based on the reduction of MTT to formazan by metabolically active cells [94]. The study determined that the methanol, petroleum and dichloromethane mixture demonstrated improved activity against TB [47]. The investigators noted that further work is necessary to identify additional compounds of polyphenols (occurring in many natural products) that are responsible for the anti-tuberculosis activity; additionally these compounds should be purified to determine important cellular targets [94].

### 4.5. Tea Polyphenols

The tea plant *Camellia sinensis* is believed to have been discovered and cultivated in Southeast Asia. Tea consumption can be traced back to 2737 BCE when, according to Chinese history, emperor Shen Nung, discovered and consumed tea for the first time [112]. Tea contains many bioactive chemicals and is rich in flavonoids, including catechins and their derivatives [113]. These polyphenolic compounds, including epigallocatechin gallate (EGCG), are believed to contribute to the beneficial effects ascribed to tea [113]. Tea polyphenols from black and green teas are being examined as possible adjuvants in vaccines [114]. They are also being studied for use in treating various medical conditions, including the prevention of cardiovascular disease, obesity and dental caries [115]. Additionally, infectious disease researchers are investigating tea polyphenols for the treatment of infectious diseases, including TB. In one Chinese study, researchers explored factors inducing the reactivation of *M. tuberculosis* in patients. A structured questionnaire was given using 63,257 subjects recruited from the Singapore Chinese Health Study. Additionally, the National Tuberculosis Notification Registry (China) was used to identify active TB cases. Assessments of tea drinking and its effects were taken from structured questionnaires and considered factors, such as body mass index, smoking status, alcohol consumption and age [99]. Consumption of tea and coffee were documented in these individuals; specifically, if individuals consumed only black tea, black and green teas, only green tea or only coffee. The subjects were followed for an average of 16.8 years (±5.2 years) to document if consumption of teas or coffee would decrease the incidence of TB. The results indicated that drinking black or green teas (in a dose-dependent manner) were inversely associated with risk of developing active TB [99]. Additionally, leaner study participants that consumed tea on a daily basis were at a reduced risk of contracting tuberculosis when compared to overweight/obese counterparts as well as weekly/daily alcohol drinkers [99]. It is important to note that there was no control group listed in the study (i.e., documenting individuals who did not consume either tea or coffee). [116].

A second study examined tea polyphenols that target the mycobacterial cell wall [100]. *Mycobacterium smegmatis* mc2 155 was incubated in Luria-Bertani (LB) broth and subjected to various concentrations of EGCG derived from green tea. Suspended mycobacterial cells were later subjected to electron microscopy, liquid chromatography-mass spectrometry (LC-MS) and high-performance liquid chromatography (HPLC)). Electron microscopy indicated structural damage to the *Mycobacterium* cell wall with increasing amounts of EGCG; authors indicated that the optimal concentration of EGCG causing bacterial cell wall damage was 20 μg/mL [100]. The content and structure of EGCG in LB media was confirmed by HPLC and LC-MS; both procedures are utilized in laboratories to isolate the individual components of a mixture [100]. A third study analyzed green tea catechins (a group of flavonoids including EGCG) and triclosan on TB organisms. Triclosan has been used for more than 30 years as a general antibacterial and antifungal agent; it inhibits lipid biosynthesis by blocking a key enzyme InhA, an enoyl–acyl carrier protein reductase [101]. InhA is integral in the mycolic acid structure of the mycobacterial cell wall. Inhibition of InhA results in a weakened mycolic acid structure and subsequent cell damage and/or death [101,102]. Scientists decided to utilize green tea along with triclosan in TB experiments; both compounds were used to target the InhA enzyme of tuberculosis [117].

In a third study, direct binding assays using [^3^H]-EGCG, fluorescence titration and docking studies were performed. EGCG was found to inhibit InhA and bind reversibly at or near the binding site of nicotinamide adenine dinucleotide (NADH). NADH is co-enzyme required for production of energy in both eukaryotic and bacterial cells. Affinity of EGCG for InhA was increased 5-fold in the presence of triclosan; affinity to triclosan was increased 8-fold in the presence of EGCG [117]. Researchers further investigated the effect of EGCG on the expression of the tryptophan-aspartate containing coat protein (TACO) gene. The TACO gene (also known as Coronin-1a) represents a coat protein on phagosomes: EGCG was found to inhibit autophagosome formation in *M. tuberculosis*. [118]. RT-PCR, reporter assay, flow cytometry and colony counts were also evaluated. EGCG was able to downregulate TACO gene transcription by inhibition of Sp1 transcriptase activity (transcriptases are important in protein synthesis). EGCG also decreased bacterial survival within macrophages [119]. It is noteworthy that triclosan is identified as toxic; there is also evidence that bacterial organisms can develop antimicrobial resistance [101]. The US FDA banned the inclusion of triclosan in compounds such as soap products, but it is still allowed in toothpastes, cosmetics and antiseptic soaps [120]. The European Union banned triclosan from all human hygiene biocidal products starting January 2017; however, triclosan is still allowed in toothpaste [120].

### 4.6. Resveratrol

Resveratrol, a stilbenoid polyphenol, is commonly found in foods such as peanuts, wine and cranberries. Resveratrol extract, taken from the roots of *Rheum rhaponticum*, also known as rhubarb, was examined for its antibacterial properties against *M. tuberculosis* and *M. bovis*. Separation of solid plant material from liquid containing resveratrol was completed by employing methanol in an ultrasonic bath; further dissolution was performed either by dissolving in DMSO or ethyl acetate and water. Cultured *M. tuberculosis* was then added to vials containing the extracted resveratrol and incubated for 7 days. It was discovered that resveratrol extracts had a significant antimycobacterial effect on both strains of mycobacterium [121]. Resveratrol and its analogues were scrutinized in another study against *M. tuberculosis*. Microbroth dilution tests were performed with varying amounts of resveratrol. Studies indicated that resveratrol completely prevented mycobacterial growth at 200 µg/mL (MIC 50–200 µg/mL) [122]. Resveratrol is known to activate abyssinone II, a prenylated flavonoid and Sirt1, a gene important in cellular regulation. This is thought to modulate the inflammatory response caused in TB [123]. Abyssinone II and Sirt1 can also be a possible target for therapy or prevention of TB.

### 4.7. Propolis

Propolis, a bee extract consisting mostly of wax and resin, is widely used in natural medicine for its therapeutic benefits [124,125]. The medicinal properties of propolis (especially antioxidant, anti-inflammatory and antimicrobial properties) have attracted the interest of investigators from many fields of study including pharmacology, botany and phytochemistry [124]. Flavonoids are the main compounds of propolis and are responsible for the principal pharmacological effects [125]. Propolis cannot be used in its’ unprocessed form; it is purified by extraction to remove the inactive material and preserve the polyphenolic fraction [126]. Ethanolic extracts of propolis (EEP) were explored to determine if the substance has synergistic effects with anti-tuberculosis medications [103]; previous work indicated EEP had a synergistic effect with antibiotics including the decreased growth of *Staphylococcus aureus* [103]. Six standard laboratory strains of *Mycobacterium* species (*M. tuberculosis* (H37Rv, HjyRa; *M. kansasii*, *M. xenopei*, *M. intracellulare* and *M. bovis*) and eight *M. tuberculosis* strains (M. 208, M. 223, M. 214, M. 241, M. 252, M. 260, M. 323 and M. 333) were isolated from patients that had been pretreated with antibiotics. Three strains of TB (M. 2563, M. 2583 and M. 2976) were isolated from patients that were not treated with antibiotics. All organisms were grown on Löwenstein-Jensen (LJ) agar containing varying concentration of antimicrobials (streptomycin, rifamycin, isoniazid and ethambutol) [103]. Variable amounts of EEP was later added to LJ plates and observed for inhibition of growth; patterns were deemed “susceptible” or “resistant” to the combined antibiotic and EEP treatment.

It was determined that combined antibiotic and EEP treatments reduced the growth rate of 14 out of 17 strains of TB. In 12 out of 17 strains, EEP enhanced sensitivity of mycobacterium to antibiotic medication [103]. Additionally, scientists noted that EPPs are strongly hydrophobic with high affinity towards lipids; lipids are abundant and distinct in the outer cell surface of *Mycobacteria*, and EPPs strongly binds to those components [103]. The bactericidal activity of EEP was associated with the virulence of *Mycobacteria*, and that when such virulence is augmented, the antimycobacterial activity of EEP is significantly increased [103].

### 4.8. Lactacystin

Lactacystin, produced by the microorganism *Streptomyces lactacystinaeus*, was initially discovered to inhibit cellular growth of mouse neuroblastoma cells [104]. Lactacystin is an irreversible proteasomal inhibitor (specifically at the 20S proteasome β-subunit) and inhibits peptide hydrolysis (at the 26S complex) [104]. Proteasomes are responsible for most of the non-lysosomal protein degradation in bacterial and eukaryotic cells [104,127]. The proteasome is thought to be necessary for the virulence of TB, since it is essential for the bacterial nitrogen metabolism. However, the degree of the proteasome’s function in bacterial survival and growth is not fully understood [128]. It is important to note that proteasome inhibitors have been found to be active in the non-replicating state of *M. tuberculosis* [129]. Thus, lactacystin could be a possible treatment for latent-TB infections and need further in-depth evaluation.

### 4.9. Fellutamide B

Peptide aldehydes, such as lipopeptide aldehyde (fellutamide B), were discovered in the marine fungus *Penicillium fellutanum*. In 2010, fellutamide B was discovered to be a potent proteasome inhibitor of *M. tuberculosis*. It inhibited both *M. tuberculosis* and human proteasomes in a time-dependent manner [105]. Kinetic measurements, high-throughput screen, progress curves, x-ray diffraction and crystallization plates were used to quantify the activity of fellutamide B. Results demonstrated that fellutamide B was a strong *M. tuberculosis* proteasome inhibitor; as a result, organisms were more susceptible to toxic nitric oxide stress [105]. Fellutamide B was shown to be over 1,000-fold more potent against *M. tuberculosis* than other peptide aldehydes and inhibited *M. tuberculosis* proteasomes with different kinetic mechanisms [105]. Researchers noted some adverse side effects to fellutamide B, including cellular and neurotoxicity [130].

## 5. Conclusions and Future Directions

*M. tuberculosis* is becoming more resistant to conventional antibiotic treatments. Natural compounds can be used to augment the actions of anti-TB drugs and possibly fill in the gaps where traditional prescription medications have become less effective. Prevention and treatment approaches along with natural compounds could possibly help reduce drug resistance and may be a practical solution. Natural compounds possess a plethora of antimycobacterial properties, and as discussed, focus on different therapeutic targets. For instance, natural compounds, such as polyphenol EGCG targets bacterial cell wall structure [119]; ethanolic extract of propolis (EEP) augments the sensitivity of mycobacterium to antibiotic treatment [131]. Additionally, curcumin nanoparticles decrease hepatotoxicity with INH treatment and shorten the duration of TB treatment time [86]. Curcumin also inhibits NF-κB, a protein complex that controls target cell survival, cytokine production and transcription of DNA. Moreover, NF-κB is important in inducing the inflammatory response of the host’s immune system causing inflammation and tissue damage [132]; inhibiting NF-κB can decrease DNA replication and proliferation of *M. tuberculosis* as well as decrease inflammation and tissue damage of the host cell. Fellutamide B, a proteasome inhibitor, may be used as adjunct treatment for tuberculosis. Catechins (phenols found in beverages such as green tea) has been demonstrated to inhibit InhA, an important enzyme involved in type II fatty acid biosynthesis [119]. It is noteworthy that many of these studies combined natural treatments with standard TB therapies.

Natural products should continue to be investigated for the treatment of active tuberculosis. It is important to note that many of the studies cited in this review have been conducted via approaches such as molecular assays, mouse models, human cells, and bacterial culture [84,86,99,108]. Additionally, investigations of large cohorts of tea and coffee drinkers [116] and those consuming Immunoxel (Dzherelo) as a preventative treatment of TB have also taken place [24]. It is imperative that these studies be replicated to determine the scientific veracity of their findings [133,134]. For example, studies regarding the herbal antimalarial compound artemisinin [77,135] and various liver diseases [136] have been highly successful. Over time, human clinical studies can also be conducted to confirm or refute hypotheses stated by those investigating natural compounds used to treat TB. This can be challenging since it is estimated that the average out-of-pocket cost per new medicinal compounds are estimated at about 1.4 billion US dollars [137]. Another issue in using natural products in TB treatment is that supply of natural products is limited by effective production methods [138]. Many natural products have intricate chemical structures that make associated chemical syntheses complicated, inefficient and costly. To counter this, new methodologies such as the emerging field of metabolic engineering and synthetic biology has enhanced the ability to genetically program microbial organisms to produce natural products [138]. Researchers note that drug discovery efforts for natural products often lag behind the discovery efforts for other products in the pharmaceutical industry [139]. Innovative work continues to be performed in areas such as targeting new proteins that are important for bacterial siderophore synthesis (central for iron sequestration); inactivating intracellular growth operons (such as the *igr* locus which encodes enzymes for the breakdown of cholesterol); and targeting energy generation of bacterial cells (interfering with the electron transport chain and inhibiting ATP synthesis) [139]. It is important to make certain that natural products are of high quality, authentic, properly formulated, consistently extracted from their sources and not contaminated with other products [140]. Novel natural compounds are continuing to be studied; it is hoped that these substances will be useful in treating tuberculosis infections.

## Figures and Tables

**Figure 1 molecules-25-03011-f001:**
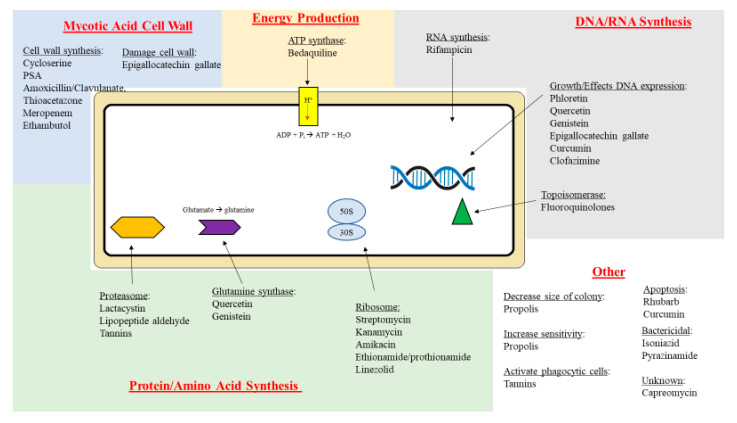
The mechanisms of different anti-TB drugs and natural compounds. It is also important to note that natural products, shown in the figure, have been used in combination with conventional anti-TB agents (see text for details and references).

**Table 1 molecules-25-03011-t001:** Drugs used for the treatment of tuberculosis [11,12,13].

Anti-TB Therapy				
**First-Line Defense** ***(Treatment for TB)***	Brand	Generic	Mechanism of Action	Year of Approval
	Nydrazid	Isoniazid *	Bactericidal agent; inhibits the enoyl-acyl carrier protein reductase InhA upon Kat activation (important for virulence); related to mycolic acid synthesis	1952
	Rifadin, Rimactane	Rifampicin *	Inhibits bacterial RNA synthesis	1971
	Myambutol	Ethambutol *	Inhibits arabinosyl transferase in bacteriostatic manner	1961
	Pyrazinamide	Pyrazinamide *	Interferes with mRNA binding Bacteriostatic/bactericidal	1972
	Streptomycin	Streptomycin sulfate *, Streptomycin nitrate	Inhibits bacterial protein synthesis	1998
**Second-Line Defense** ***(Treatment for resistant TB)***	Group A (*Fluoroquinolones*):			
	Levaquin, Quixin, Iquix	Levofloxacin	Interferes with topoisomerase IV and DNA gyrase (DNA replication)	1996
	Avelox, Vigamox, Moxeza	Moxifloxacin	Inhibits topoisomerases II and IV (DNA replication)	1999
	Tequin, Zymar, Zymaxid	Gatifloxacin	Inhibits DNA topoisomerases II and IV as well as DNA gyrase (DNA replication)	1999
	Group B *(Injectable Antibiotics):*			
	Amikin	Amikacin	Aminoglycoside active against susceptible Gram-negative pathogens and Gram-positive bacteria	1976
	Capastat sulfate	Capreomycin *	Cyclic polypeptide antimicrobial	1968
	Kanamycin A	Kanamycin	Binds to the bacterial 30S ribosomal subunit-inhibits protein synthesis)	1981
	Group C (*Oral drugs*):			
	Trecator/Prothionamide	Ethionamide */Prothionamide	Inhibits mycolic acid synthesis	1965
	Seromycin	Cycloserine *	Inhibits cell wall synthesis	1964
	Paser	*para*-Aminosalicylic acid	Inhibits folic acid synthesis, inhibits cell wall synthesis	1994
	Lamprene	Clofazimine	Inhibits mycobacterium growth and binds to mycobacterial DNA	1986
**Third-Line Defense** ***(Treatment for MDR TB)***	Sirturo	Bedaquiline *	Inhibits mycobacterial ATP synthase	2012
	Zyvox	Linezolid	Inhibits bacterial reproduction of aerobic Gram-positive bacteria, some Gram-negative bacteria, and anaerobic bacteria; inhibits protein synthesis by binding to bacterial 23s ribosomal RNA of the 50s subunit	2000

* Indicates US FDA-approved treatments.

**Table 2 molecules-25-03011-t002:** Natural compounds with potential for the treatment of TB tuberculosis.

Natural Compound	Source	Chemical Constituent	Activity	References
*Curcuma longa*	Turmeric	Curcumin	Inhibits activation of NF-κB, and caspase-3; enhances T cell- mediated immunity and prevents post therapy susceptibility to reinfection/reactivation of Mtb.	[84,85,86]
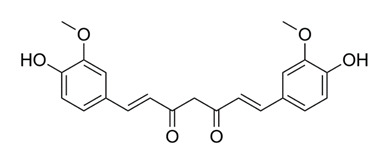				
*Malus domestica*	Apples (peels), apple leaves	Phloretin	Inhibits release of TNF-α, IL-1β and IL-12 and reduces phosphorylation of JNK, ERK and p38 MAPK.	[87]
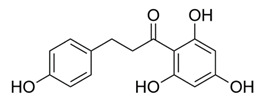				
*Euphoria paralias*	Whole plant extract	Quercetin	Inhibits glutamine synthetase and isocitrate lyase (important in nitrogen and TCA cycle, respectively).	[88,89,90,91,92]
*Punica granatum*	Pomegranate juice and peel	Quercetin		

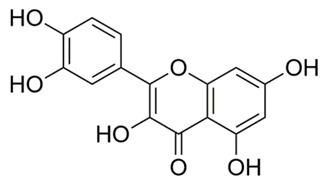				
*Glycine max*	Soybeans	Genistein	Impairs B7-1 expression and nitric oxide release in MTSA-transfected macrophages.	[93]
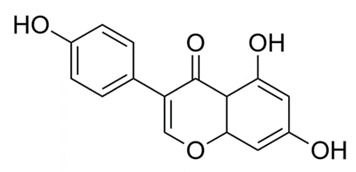				
*Globularia alypum* L.	Perennial flowering plant	Tannins	Activates phagocytic cells, proteasome inhibitor.	[94,95,96,97,98]
*Camellia sinensis* (L.) Kuntze	Green and black tea	Epigallocatechin gallate	Downregulates TACO gene transcription, inflicts structural damage to Mycobacterial cell wall, by inhibiting InhA.	[99,100,101,102]
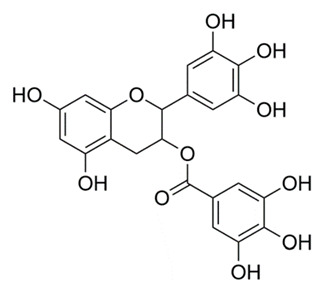				
*Rheum rhaponticum*	Rhubarb	Resveratrol extracts	Increases cell apoptosis.	[103,104]
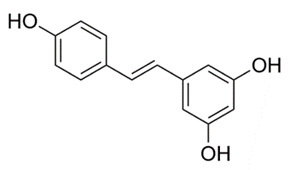				
	Honeybee hives	Ethanolic extract of propolis	Decreases the size and number of Mtb colonies and increases sensitivity of mycobacteria to antibiotic therapy.	[103]
*Streptomyces lactacystinaeus*	Microorganism	Lactacystin	Inhibits proteasome function.	[104]
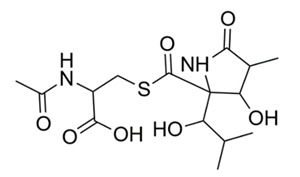				
*Penicillium fellutanum*	Marine fungus	Lipopeptide aldehyde (Fellutamide B)	Inhibits Mtb proteasome and human proteasome β5 function.	[105]
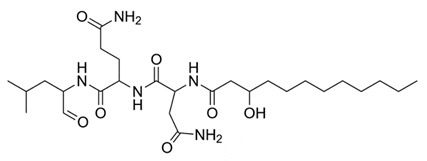				

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
