# Peer review of "The War against Tuberculosis: A Review of Natural Compounds and Their Derivatives"

_molecules, 2020, doi:10.3390/molecules25133011_

Round 1

Reviewer 1 Report

The present article aims to review the advance in TB therapy, with a particular focus on natural compounds. The topics is certainly of great interest, but the review is quite difficult to read, and presents several flaws.

First of all, it seem quite unbalanced, as only 1 paragraph out of four really deal with natural compounds, while more than half of the review present several aspects of TB, that although of importance, appear redundant or out of topic (for instance the description of the diagnostic approaches, or the long parts regarding the treatments), that should be avoided or considerably reduced, in favour of a more articulated discussion of natural compounds.

The paragraph about the different natural compounds appears mainly as a list of the different studies, but lacks a real discussion about the mechanism of action (known or hypothesized), the current status and the true potential of these compounds. By contrast, the description of the studies presents too much technical details about the experimental approaches, that contributes to make difficult the reading.

Several other minor points are highlighted in the attached file.

Author Response

The authors of this manuscript express their sincere thanks to the reviewer for the critical assessment of our work. The authors have acted upon the recommendations of the reviewer which have resulted in a significant enhancement of the quality of this manuscript. All modifications incorporated in the manuscript are highlighted using red color font. A “point-by-point” response to the reviewers’ comments is outlined below.

Comment 1:

The present article aims to review the advance in TB therapy, with a particular focus on natural compounds. The topics is certainly of great interest, but the review is quite difficult to read, and presents several flaws.

Response:

We sincerely thank the reviewer for his/her expertise and time and effort towards reading our manuscript with excellent suggestions. The reviewer’s comments and suggestions have been very helpful in revising our manuscript as presented below.

Comment 2:

First of all, it seem quite unbalanced, as only 1 paragraph out of four really deal with natural compounds, while more than half of the review present several aspects of TB, that although of importance, appear redundant or out of topic (for instance the description of the diagnostic approaches, or the long parts regarding the treatments), that should be avoided or considerably reduced, in favor of a more articulated discussion of natural compounds.

Response:

We are in absolute agreement with the reviewer. More information regarding natural compounds has been added and other areas are minimized/condensed considerably in section 1 (Introduction), section 2 (Therapeutic Targets) and section 3 (Treatment) including subsections.

Comment 3:

The paragraph about the different natural compounds appears mainly as a list of the different studies but lacks a real discussion about the mechanism of action (known or hypothesized), the current status and the true potential of these compounds. By contrast, the description of the studies presents too much technical details about the experimental approaches, that contributes to make difficult the reading.

Response:

More information on natural compounds and mechanisms of action has been added. For example, we provided critical information regarding tea polyphenols and the mycolic acid wall (page 14; lines 456-460; 463-464). Another area was lactacystin (section 4.8; page 15; lines 520-526) where we have added discussion on its mechanism of action with special emphasis on proteasomal inhibition. We have also made various modifications and refinements throughout the manuscript and all these changes are highlighted in red color.

Reviewer 2 Report

The authors designed a good aim to the manuscript, focusing in tuberculosis,  a recurrent and concerning problem of public health, with a growing antibacterial resistance profile;

The references used were adequated and the time period of the bibliographical search are adequated also;

The text is well writen and its construction is clear and with easy comorehension.

However, I detected some mispellings and few grammatical erros in the text. However, even this little problem do not compromisse the comprehension.

By this fact, I recommmend the ACCEPTANCE OF THE MANSUCRIPT and I believe that the english correction made by the editorial office from MDPI is enough to correct the english.

Author Response

The authors of this manuscript express their sincere thanks to the reviewer for the critical assessment of our work. The authors have acted upon the recommendations of the reviewer which have resulted in a significant enhancement of the quality of this manuscript. All modifications incorporated in the manuscript are highlighted using red color font. A “point-by-point” response to the reviewers’ comments is outlined below.

Comment 1:

The authors designed a good aim to the manuscript, focusing in tuberculosis,  a recurrent and concerning problem of public health, with a growing antibacterial resistance profile;

The references used were adequated and the time period of the bibliographical search are adequated also;

The text is well writen and its construction is clear and with easy comorehension.

Response:

We are encouraged by the reviewer’s generous comments and appreciation of our work.

Comment 2:

However, I detected some mispellings and few grammatical erros in the text. However, even this little problem do not compromisse the comprehension.

Response:

We sincerely thank the reviewer for a careful reading of our manuscript. Our manuscript has been subjected to a thorough editing with the elimination of spelling and grammatical errors.

Comment 3:

By this fact, I recommmend the ACCEPTANCE OF THE MANSUCRIPT and I believe that the english correction made by the editorial office from MDPI is enough to correct the english.

Response:

We are encouraged by the reviewer’s recommendation for the acceptance of our manuscript. We sincerely believe we have tried our best to make all the necessary corrections.

Reviewer 3 Report

This review by Shipp et al., provides a useful and comprehensive review of the literature relating to the use of natural products in the treatment of TB. I have only minor comments:

  • While the English used throughout is generally acceptable, there are certain paragraphs (such as the last paragraph of introduction) that need attention.
  • Care should be taken regarding the definition of acronyms, for example: "RIF" is used prior to its definition. 
  • Addition of references to tables would be useful to readers looking to easily access the primary studies. 

Author Response

The authors of this manuscript express their sincere thanks to the reviewers for the critical assessment of our work. The authors have acted upon the recommendations of the reviewers which have resulted in a significant enhancement of the quality of this manuscript. All modifications incorporated in the manuscript are highlighted using red color font. A “point-by-point” response to the reviewers’ comments is outlined below.

Comment 1:

This review by Shipp et al., provides a useful and comprehensive review of the literature relating to the use of natural products in the treatment of TB. I have only minor comments.

Response:

We sincerely thank the reviewer for his/her expertise and time and effort towards reading our manuscript with excellent suggestions. The reviewer’s comments and suggestions have been very helpful in revising our manuscript as presented below.

Comment 2:

While the English used throughout is generally acceptable, there are certain paragraphs (such as the last paragraph of introduction) that need attention.

Response:

We have checked the entire manuscript and made several additions and changes. The introduction section has been revised as suggested.

Comment 3:

Care should be taken regarding the definition of acronyms, for example: "RIF" is used prior to its definition. 

Response:

We have used the full form of all abbreviations used for the first time.

Comment 4:

Addition of references to tables would be useful to readers looking to easily access the primary studies. 

Response:

We think the reviewer has made an excellent suggestion. We have included references in Table 2.

Round 2

Reviewer 1 Report

The revised version of the review is now suitable for pubblication.